# Trending Towards Safer Breast Cancer Surgeries? Examining Acute Complication Rates from A 13-Year NSQIP Analysis

**DOI:** 10.3390/cancers11020253

**Published:** 2019-02-21

**Authors:** Michael M. Jonczyk, Jolie Jean, Roger Graham, Abhishek Chatterjee

**Affiliations:** 1Department of Surgery, Tufts Medical Center, 800 Washington Street, South Building, 4th Floor, Boston, MA 02111, USA; rgraham@tuftsmedicalcenter.org (R.G.); achatterjee1@tuftsmedicalcenter.org (A.C.); 2Department of Clinical and Translational Science, Tufts University Sackler Graduate School, 136 Harrison Ave #813, Boston, MA 02111, USA; 3Tufts University School of Medicine, 145 Harrison Ave, Boston, MA 02111, USA; jolie.jean@tufts.edu

**Keywords:** breast conservation surgery, oncoplastic surgery, mastectomy, mastectomy with reconstruction, complication rate, comorbidity, trend analysis

## Abstract

As breast cancer surgery continues to evolve, this study highlights the acute complication rates and predisposing risks following partial mastectomy (PM), mastectomy(M), mastectomy with muscular flap reconstruction (M + MF), mastectomy with implant reconstruction (M + I), and oncoplastic surgery (OPS). Data was collected from the American College of Surgeons NSQIP database (2005–2017). Complication rate and trend analyses were performed along with an assessment of odds ratios for predisposing risk factors using adjusted linear regression. 226,899 patients met the inclusion criteria. Complication rates have steadily increased in all mastectomy groups (*p* < 0.05). Cumulative complication rates between surgical categories were significantly different in each complication cluster (all *p* < 0.0001). Overall complication rates were: PM: 2.25%, OPS: 3.2%, M: 6.56%, M + MF: 13.04% and M + I: 5.68%. The most common predictive risk factors were mastectomy, increasing operative time, ASA class, BMI, smoking, recent weight loss, history of CHF, COPD and bleeding disorders (all *p* < 0.001). Patients who were non-diabetic, younger (age < 60) and treated as an outpatient all had protective OR for an acute complication (*p* < 0.0001). This study provides data comparing nationwide acute complication rates following different breast cancer surgeries. These can be used to inform patients during surgical decision making.

## 1. Introduction

Breast cancer surgery is adapting to rising patient preferences for breast reconstructive procedures. Treatment of breast cancer surgery can be classified into two overall groups: breast conserving therapy (BCT) including partial mastectomy (PM) and oncoplastic surgery (OPS), and mastectomy (MAST) including mastectomy alone (M) and M with breast reconstruction (M + R). From 2005 to 2017, the use of breast reconstruction significantly increased compared to other types of breast cancer surgery for patients with both ductal carcinoma in situ (DCIS) and invasive carcinoma (IvBC) [1]. Known for its extensive use of tissue mobilization and re-arrangement to ensure optimal reshaping with breast cosmesis [2], the use of OPS (a form of breast reconstruction) has doubled from 2% to 5%. In parallel, mastectomy with implant placement (M + I) increased from 11% to 21% [1]. Meanwhile, mastectomy with muscular flap reconstruction (M + MF) has actually declined from 4.5% to only 1% [1]. Accompanying these changes in breast reconstruction is the steady decline of non-cosmetic procedures like traditional mastectomies (M). These shifting trends in breast cancer treatment are multifactorial and are likely attributed to changes in complication rates, comorbidities, patient demographics, patient surgical preference, and oncological guidelines for appropriate surgical resection [1].

Breast reconstruction (OPS or M + R) offers patients an improved quality of life by providing an aesthetically symmetric breast together with higher patient satisfaction [2,3,4]. Unfortunately, complication rates persist in breast cancer surgery and vary (2–40%) with the type of reconstruction. They also vary depending on whether we measure short-term or long-term outcomes [2,5,6,7,8,9,10,11,12,13]. Fortunately, mortality in breast cancer surgery remains very low (<1%) regardless of the type of surgery offered [14]. Post-operative complications are influenced by multiple risk factors that surgeons should consider. Wound infections and postoperative infectious complications have been associated with smoking, prior radiation, obesity and diabetes [5,8].

Several studies have also shown that multiple comorbidities and higher American Society of Anesthesiologist (ASA) classification predict higher complication rates following all surgery subtypes. Nevertheless, specifically pertaining to breast cancer intervention, many patient factors and surgical predictors thought to influence acute postoperative complications are unknown or controversial [5,8,15,16,17]. Single institutional studies and prior reviews have generally suffered from small sample size and have lacked the power to adequately analyze the multiple variables influencing post-operative acute complications following breast cancer surgery.

Understanding surgical complications is crucial to patient safety and improving health care outcomes. Therefore, this study sought to examine the acute postoperative complication rates in breast cancer patients who underwent PM, M, M + I, M + MF, and OPS. Using the NSQIP database, we aimed to expand our understanding of predictive factors associated with different surgical procedures performed between 2005 and 2017 and evaluated trends over time.

## 2. Methods and Materials

This study follows the same methodology, inclusion/exclusion criteria, data collection and surgical categorization used in Jonczyk et al. [1]. A retrospective cohort analysis was conducted using the ACS-NSQIP database from 2005 to 2017. All participant user files (PUF) were obtained and approved by ACS NSQIP. The Institutional Review Board deemed this study exempt from institutional review given that the ACS NSQIP database is a de-identified data set.

### 2.1. Data Collection

Inclusion criteria for this study were women with classified post-operative diagnosis of invasive breast cancer (IvBC) or ductal carcinoma in-situ (DCIS) who underwent any BCT or any MAST procedure. Post-operative diagnosis was classified according to International Classification of Diseases Ninth Revision (ICD-9) code for IvBC (ICD-9, 174) or DCIS (ICD-9, 233). After October 2015, ICD Tenth Edition replaced the previous system of classification, and patients with IvBC or DCIS were classified under the appropriate ICD-10 codes: D05, D5.1-D05.99 (DCIS), and IvBC (C50). In order to examine complications for specific interventions, each surgical group (except for PM) was further divided into categories (CG) shown in Appendix A. A schematic of surgical (M, PM, OPS, M + R) categorization using CPT codes is shown in Appendix A. Exclusion criteria included males, surgery for benign breast disease, lobular carcinoma in situ, patients undergoing breast cancer surgery with 2 CPT codes with ambiguous category placement and septic patients at time of surgery.

### 2.2. Complications and Outcome Measures

We identified 16 acute complications in the NSQIP database that were collected prospectively in a 30-day post-operative period. We used these complications and clustered them into eight groups based on their medical similarity. Table 1 depicts complication clustering and the individual complications included.

Demographics, patient comorbidities, and surgical factors were also collected for each surgical category (Table 2). Body mass index (BMI) was not included in NSQIP and was therefore calculated using weight (lbs.) divided by squared height, multiplied by 703 [13]. Each patient was categorized as: underweight (BMI < 18.5), normal (BMI 18.5–25), overweight (BMI 25–30) or obese (BMI > 30).

### 2.3. NSQIP Variable Definitions

NSQIP defines patients at risk for bleeding due to any condition with deficiency of clotting elements (Vitamin K deficiency, hemophilia’s, thrombocytopenia, or on chronic anticoagulation other than aspirin). Chemotherapy and radiation were defined as being administered pre-operatively for malignancy in less than 30 days and 90 days, respectively. Open wound infections (OWI) were any breach of skin integrity with or without cellulitis or purulent exudate when leaving the operating room and this included the use of drain devices or negative pressure wound devices. OWI did not include scabbed over wound or Band-Aid covered sores (break in skin), tracheostomy, oral sores and ostomies. For this analysis, we associated OWI with drain placement or wound vac placement. Recent weight loss was defined as greater than 10% unintentional loss of body weight. Hypertension (HTN) had to be documented, and patients had to be on medication for over 2 weeks prior to surgical intervention.

### 2.4. Statistics

All analyses were performed using R-Studio software. Chi-square tests analyses were performed for demographic and complication rate analysis. Smoothed linear regression was used to adjust a best fit line and then a non-parametric Mann- Kendall test was used to assess complication temporal trends. The variables in Table 3 were used as covariates for a stepwise logistical regression model for each clustered complication grouping.

Each covariate was analyzed for significance in a univariate logistical regression (*p* < 0.05) for each clustered complication. All significant covariates were used in a multivariable logistical regression (MLR) to calculate the adjusted odds ratio (OR) for acquiring each complication. Each covariate was compared to its baseline covariate and a computed OR predicts an association to the baseline covariate. Patient baseline covariates used in the MLR were as follows: diabetes treated with IV insulin, white race, non-smokers and no prior smoking pack-years, older age (>60 years old), pathology (DCIS), normal BMI, no PMH of bleeding disorder, renal failure, angina, CHF, HTN, or COPD, and no recent weight loss. Surgical baseline covariates used in MLR were as follows: inpatient admission status, ASA class 1, operative time < 1 h, surgery (PM), no open wound infection, and no prior operation within 30 days. All results were considered significant at *p* values < 0.05 level.

## 3. Results

### 3.1. Participant Pool and Demographics

Between 2005 and 2017, over 6 million patients were included in the NSQIP database and roughly 300,000 patients underwent breast cancer surgery. A total of 226,899 (77.9%) women met our inclusion criteria for the present analysis (Appendix A). Demographics, patient comorbidities, and surgical factors were significantly different (<0.001) among all five groups (Table 2). Compared to other groups, the M + I group had the highest incidence of younger patients and a lower incidence of the following preoperative indicators: lower ASA class, fewer open wound infections or systemic infections within 48 hours of surgery, and fewer prior operations in the last 30 days.

### 3.2. Overall Complication Rate and Trend Analysis

Cumulative complication rates were analyzed between 2005–2017 and clustered into their appropriate complication groups for each surgical intervention (Table 4).

All complication clusters were significant (<0.0001). The overall complication rates per surgical intervention were as follows: PM 2.25%, OPS 3.2%, M 6.56%, M + MF 13.04% and M + I 5.68%. Wound, infectious, respiratory, bleeding and thromboembolic complications were highest in the M + MF group. The M group had the highest rates of renal, cardiac and stroke complications. Table 5 and Figure 1 depict the 13-year adjusted smoothed trend analysis in nationwide breast cancer surgeries.

From 2005 to 2017, there was no significant trend change for acute postoperative complications in all patients in the BCT group (PM and OcPs); (*p* >0.05). However, all categories in the MAST group had increased trends for complication rates as follows: M category 5.4% to 6.8% (*p* = 0.004), M + I 5.5% to 6.1% (*p* = 0.02) and M + MF 7.4% to 13.3% (*p* = 0.01).

### 3.3. Independent Factors Associated with Complications (p < 0.05)

Factors in the unadjusted analysis most likely associated with any complication were: outpatient surgery, increased BMI, smoking, M + R, OPS, ASA Class, increasing operative time, PMH of diabetics, renal failure, angina, CHF, COPD and HTN. Factors analyzed that were least likely to be associated with any cluster of complications included: pathology of the cancer, use of chemotherapy, PMH of dyspnea, angina or renal failure, prior operation in the last 30 days, pregnancy and recent pneumonia within 48 h. Unadjusted OR can be seen in Appendix AA–C, and significant covariates were used in the multiple linear regression (MLR). Table 6 and Table 7 show MLR analysis of predictive factors associated with patient characteristics and surgical predictors, respectively.

### 3.4. Adjusted Predictive Factors

#### 3.4.1. Predicting Factors Associated with Lower Complication Rates

Complications had a lower associated risk when post-operatively treated as outpatients (decrease risk ranging from 16% to 87%) when compared to inpatients (*p* < 0.03). Compared to diabetics on insulin therapy, diabetics on oral medication were 25% less likely to acquire wound complications and had a 26% lower incidence of other infections (*p* = 0.005; *p* < 0.02). Likewise, non-diabetics reduced their odds of wound complications by 40%, infections by 41%, bleeding complications requiring transfusions by 45%, and renal complications by 67% (*p* < 0.004). Younger patients (<60 years of age) were 47% less likely to have cardiac complications (*p* = 0.02), 39% less likely to acquire respiratory complications, and had a 67% lower risk for stroke. 

#### 3.4.2. Predicting Factors Associated with Higher Complication Rates

Smokers had a 1.6× higher odds of wound complications and 1.2× higher risk for infectious complications (*p* < 0.03). Compared to patients with a normal BMI, increasing BMI (obese and overweight patients) correlated with more complications. Obese patients were more likely to have wound complications by a factor of 2.2, infections by a factor of 1.68, respiratory complications by a factor of 1.5, and thromboembolic complications by a factor of 1.66 (all *p* < 0.03). However, underweight BMI patients had an increased risk of respiratory (2.22×) and cardiac (2.83) complications (both *p* < 0.2). Preoperative unintentional weight loss was associated with increased infections by a factor of 3.46 (*p* = 0.04) and respiratory complication by a factor of 2.7 (*p* = 0.04). Steroidal use was associated with increased bleeding risk by a factor of 1.82 (*p* = 0.001).

Comorbidities correlated to an increased risk for five complication clusters: wound, infection, respiratory, bleeding and renal. COPD predicted a higher risk of wound complications by 1.29× (*p* = 0.003), infections by 1.36× (*p* = 0.02) and respiratory complication by 1.81× (*p* = 0.007). CHF increased odds of bleeding complications 3-fold (*p* = 0.0001) and renal complications 5-fold (*p* = 0.004). PMH of angina was associated with a two-fold risk of wound complications (*p* = 0.003) and cardiac complications risk 5.20× (*p* = 0.03). Hypertensive patients on medication were more likely to acquire infections by a factor of 1.2× (*p* = 0.005) and cardiac complications by a factor of 2.3× (*p* = 0.003). Patients were 10× more likely to have renal complication when having a PMH of renal failure (*p* = 0.03). Similarly, PMH of a bleeding disorder was associated with a two-fold odds of bleeding complications (<0.0001), respiratory compromise (<0.0001), infections (*p* = 0.0006) and renal complications (*p* = 0.03).

Perioperative surgical predictors were notable for an associated trend in numerous covariate categories. Overall, all MAST procedures were almost twice as likely to suffer from wound, infectious, bleeding, thromboembolic and neurological complications (*p* < 0.002). Of note, M + MF was a particularly significant risk factor for predicting complications. There was a 10-fold increase for bleeding complications in the M + MF group (<0.0001). A trend correlating increase in operative time was associated with increasing wound complications by 1.32–2.79× (*p* < 0.0001), infections 1.34–2.55× (*p* < 0.01), bleeding complications 2.9–7.16× (*p* < 0.0001) and thromboembolic complications 1.95–6.36× (*p* < 0.0001). Compared to surgical operative times of less than 1 h, operative times between 1 and 5 hours lowered the likelihood of cardiac complications by 43–73% and risk of stroke by 70% in some women. Rising ASA class 2–4 correlated with increased likelihood of acquiring a wound complications (*p* < 0.02) and infections (*p* < 0.004).

## 4. Discussion

The incidence of breast cancer in the United States continues to increase. Therefore, it has become increasingly important to address the complication rates resulting from unique patient demographics and comorbidities [18]. In 1998, the Women’s Health and Cancer Rights Act (WHCRA) offered patients protection and insurance coverage for reconstructive breast intervention following a mastectomy [19]. Since then, reconstructive rates have increased dramatically. With this rise in breast reconstruction, studies outlining the risk factors associated with these modern surgeries have become a significant part of the surgical decision making process. To our knowledge, our study is the largest analysis of surgical trends with acute post-operative complications in all breast cancer interventions in context to IvBC and DCIS.

Our data analysis shows no overall complication trends in patients undergoing BCT. However, MAST procedures all have increasing trends for complication rates and M + MF had the highest complication rate in the majority of complication clusters. The diminishing use of M + MF may be the result of its prolonged operative time and higher acute complication rate. Within the MAST group, M + I had the lowest overall complication rate, even when compared to M alone. This is possibly attributed to its use in healthier patients as seen in patient demographics (Table 2). Compared to previous single institution analyses, our data falls within previously described ranges of early or acute post-operative complications rates ranging as follows: PM 1.2–1.4% [20,21], OPS 4.8–20% [16,20,21,22,23], M 6% [14], M + I 4–40% [6,8,24], M + MF 16–23.7% [5]. Varying with the type of reconstructive procedure, long-term complication rates (>30 days) were typically higher [6,9,22,24,25]. In our data, wound complications and infections represented the majority of overall complication rates in all surgical categories. Bleeding complications requiring transfusions were the third most common complication. The occurrence of a bleeding complication was 4.34% in M + MF, followed by 1.2% in M, 0.63% in M + I and 0.38% in OPS.

Overall, our analysis relates to and expands on previously published data demonstrating that patients undergoing OPS had the lowest complication rates when compared to patients undergoing M alone or M with any reconstructive procedure [20,21,23,26,27,28]. OPS is gaining popularity due to its high aesthetic satisfaction, increased tumor free margin rate, and decreased recurrence rate when compared to other surgical interventions [1,15,16,22,26,29]. The lower complication rate offers surgeons one more reason to offer OPS. For patients with large tumors located in the upper inner or lower poles of the breast, OPS is now frequently the recommended option [15]. Conversely, mastectomy procedures encompass extensive tissue removal with the added difficulties of reconstruction and skin expansion. Previously reported analyses, specific for M + R, have found associated higher complication rates (up to 40%) including early infection, bleeding, wound dehiscence, scar formation, nipple loss, capsular contracture, flap loss and functional impairment [5,6,7,8].

Similar to previously conducted retrospective studies, our data demonstrates increased acute complications rates associated with smoking [5,8,14,25,30,31], obesity [5,8,14,25,30,31] and advancing age [5,8,10,31]. Our adjusted linear regression analysis found that, when compared to diabetics treated with insulin, non-diabetics and diabetics on oral medication only had a much lower incidence of multiple complications. This is in conflict with other retrospective analyses showing no significant difference between patients with and without glycemic control [8,25,32]. Similarly, advancing age is a controversial factor for predicting acute complication rates. When adjusted for confounding health status and comorbidities, elderly patients have an increased likelihood to acquire a complication. These results were consistent with other studies [8,10,25,30]. Radiation and chemotherapy were significant predictors in the univariate analysis for multiple complications; however, in adjusted MLR, they were insignificant confounders showing no risk of acute postoperative complication [33].

Our data showed that extended operative times correlated with an additional 25% per hour increased likelihood of complications [8,14,31,34]. Likewise, increasing ASA class doubled or tripled the likelihood for complications, especially wound complications and infections [8,14,31,35]. Although surgeons consider comorbidities to be predictive of complications, there is minimal reliable breast cancer research on how certain patient factors including recent weight loss, type of surgical intervention, PMH of CHF, angina, HTN, COPD, renal failure, and bleeding disorders, affect health outcomes.

In the context of elective surgery, COPD, CHF and recent weight loss have been related to an increased risk of pulmonary complications [36]. Intuitively, cardiovascular comorbidities such as hypertension (HTN), angina and CHF are associated with higher cardiac complications [37,38]. Regarding readmission status following elective surgery, a two-fold increase has been associated with CHF [39]. Our research closely parallels the results of previously published data on several risk factors associated with non-cardiac surgeries [36,37,38,39]. For example, HTN increased odds two-fold (*p* = 0.003) for cardiac complications as did angina 5× (*p* = 0.03) and underweight BMI 3× (*p* = 0.02). Furthermore, wound and infectious complications nearly doubled in patients with comorbidities such as bleeding disorders, COPD, HTN, CHF and angina. Similarly, PMH of renal failure exaggerated renal complications ten-fold (*p* = 0.03) as did other comorbidities such as CHF, bleeding disorders and recent weight loss (albeit to a lesser extent). The same comorbidities also increased the likelihood of bleeding complications.

Identifying and quantifying these comorbidities preoperatively may allow for better stratification of patient risk and better matching of patients with different operative procedures in order to lower post-operative morbidity [40]. It is logical to adhere to published guidelines on the classification and management of comorbid illnesses prior to surgery: optimizing COPD patients according to GOLD guidelines [41,42], cardiovascular optimization of HTN, CHF, and angina according to the 2014 ACC/AHA guidelines [43], and renal management according to guidelines from the Kidney Disease: Improving Global Outcomes (KDIGO) organization or American Society of Nephrology in order to minimize acute kidney injury post-operatively [44,45,46].

Return to the OR is a factor in large part under a surgeon’s control, done primarily for close/positive surgical margins or due to complications (current rate = 5%; see Table 2). Following this review, and with enhanced knowledge of factors predictive of complications, appropriate action can be taken to further diminish them. Our predictors should allow surgeons to better consent patients and identify those at particular risk for specific complications. Patients with any drains or wound vacuum device placement are more likely to be at risk for a post-operative complication and should be counseled preoperatively We should always try to keep operative time to a minimum (when feasible) and recommend treating patients in the outpatient setting, as both factors were associated with fewer complications across all category types. Interestingly, despite the additional surgery, OPS was associated with the lowest complication rate (other than PM alone), and this factor should be considered in determining the optimal surgical approach for any patient considering mastectomy or mastectomy with reconstruction.

There were several limitations in this study. Oncologic factors, such as tumor size, preoperative nodal involvement, and stage, were not included in the dataset, thereby precluding us from determining their role in deciding the choice of surgical intervention. Interpretation of NSQIP database based on appropriate coding can be problematic, especially with the growing number of NSQIP and CPT codes for OPS. OPS as a reconstructive technique has gained popularity, but CPT codes may vary significantly from one institution to another. We used a coding protocol similar to one used at our institution—as OPS has no specific individual CPT code. Also, long term complications were not recorded in the NSQIP database, thereby potentially skewing the final data. Finally, certain comorbidities such as bleeding disorder and CHF may correlate with one another, but NSQIP did not provide specific medications like the purpose of blood thinners that may have affected these risks and rates of complications. This study highlights the on-going need for further, prospective studies that include the exact surgical procedure, patient comorbidities, and the concomitant use of chemotherapy and radiation therapy in order to help surgeons identify high-risk patients and lower postoperative complications.

## 5. Conclusions

As surgical interventions evolve according to oncological guidelines, patient preferences and modern reconstructive surgeries, the importance of determining and acknowledging complication rates is critical for every surgeon. This paper summarizes most of the risk factors and complications associated with the different kinds of breast surgery. While PM alone has the lowest complication rate (with positive margin rate not included as a complication), it is interesting to note that OPS offers both a form of breast reconstruction and a lower complication rate than either mastectomy alone or mastectomy with reconstruction. This factor should be taken into consideration when counseling patients who require more than a simple partial mastectomy.

## Figures and Tables

**Figure 1 cancers-11-00253-f001:**
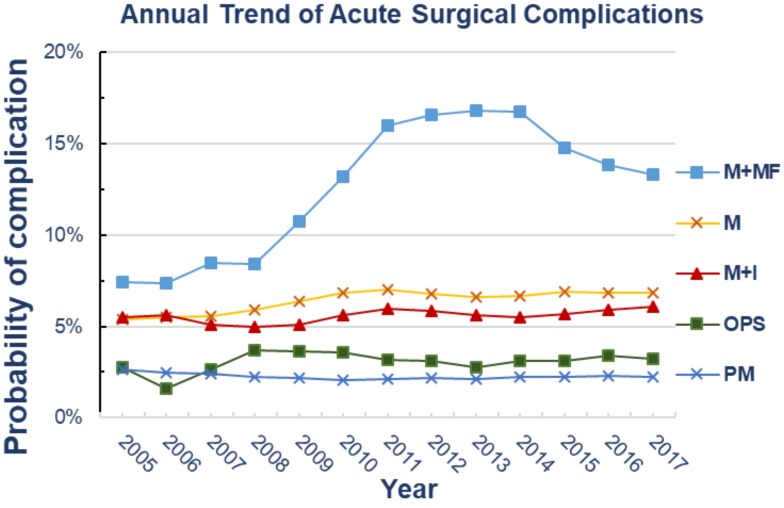
Annual trend analysis showing complication rate within each surgical category. PM: Partial Mastectomy; OPS: Oncoplastic Surgery; M: Mastectomy; M + MF: Mastectomy with Muscular Flap reconstruction; M + I: Mastectomy with Implant placement.

**Table 1 cancers-11-00253-t001:** Complication Clustered from NSQIP database.

Clusters	Individual Complication with NSQIP Code
Wound Complications	Superficial Incisional Infection SSI: SUPINFEC Deep incisional Infection DSI: WNDINFD
Infectious Complications	Organ/Space SSI: ORGSPCSSI Urinary Tract Infection UTI: URNINFEC Sepsis: OTHSYSEP Septic Shock: OTHSESHOCK
Respiratory Complications	Pneumonia: OUPNEUMO Unplanned re-intubation: REINTUB
Thromboembolic Complications	Pulmonary Embolism: PULEMBOL DVT requiring Therapy: OTHDVT
Bleeding Complications	Intraoperative or post-operative transfusion: OTHBLEED
Renal Complications	Postoperative Renal Failure: OPRENAFL Progressive Renal Insufficiency: RENAFAIL
Cardiac Complications	Cardiac arrest requiring CPR: CDARREST Myocardial infarction: CDMI
Stroke Complications	Stroke: CNSCVA

**Table 2 cancers-11-00253-t002:** Patient Demographics and Comorbidities in each Surgical Category.

Demographics (%)		PM *n* = 95,468	OPS *n* = 7279	M *n* = 70,616	M + MF *n* = 4747	M + I *n* = 44,093	*p*-Value
Age	Young: <60	56.63	54.71	42.19	75.27	78.25	<0.0001
	Older: >60	43.37	45.29	57.81	24.73	21.75	
Race	White	72.66	74.91	68.73	73.98	76.77	<0.0001
	Black	10.71	11.84	11.83	12.79	7.99	
	Asian/Pacific	3.88	4.09	6.13	3.71	4.39	
	Native	0.50	0.16	0.72	0.21	0.22	
	Unknown	12.24	8.98	12.59	9.31	10.63	
Pathology	Invasive	78.46	81.04	86.58	78.83	80.29	<0.0001
	DCIS	21.54	19.10	13.42	21.17	19.71	
Admission Status	Inpatient	7.15	15.11	51.81	93.66	61.99	<0.0001
	Outpatient	92.85	84.89	48.19	6.34	38.01	
BMI	Underweight	2.09	1.36	2.80	1.18	2.29	<0.0001
	Normal	27.27	27.17	29.20	30.90	41.27	
	Overweight	30.55	29.43	29.35	33.85	28.72	
	Obese	40.07	29.43	38.65	34.02	27.71	
Diabetic	Non-Diabetic	86.91	88.68	84.24	93.22	94.60	<0.0001
	Diabetic-Insulin	3.67	3.20	4.94	1.85	1.38	
	Diabetic-Oral	9.42	8.12	10.82	4.93	4.02	
Pregnancy Status		0.03	0.00	0.06	0.04	0.02	<0.0001
Steroid Use		1.87	1.91	2.70	1.56	1.76	<0.0001
Prior Chemotherapy		0.83	1.20	3.37	2.53	1.66	<0.0001
Prior Radiation Therapy		0.08	0.04	0.26	0.34	0.09	<0.0001
Recent Weight Loss		0.27	0.34	0.72	0.40	0.29	<0.0001
PMH of Angina		0.07	0.07	0.11	0.04	0.02	<0.0001
PMH of Hypertension		0.47	0.40	0.50	0328	0.24	<0.0001
PMH of CHF		0.27	0.29	0.43	0.06	0.04	<0.0001
PMH of Renal Failure		0.03	0.03	0.04	0.00	0.01	0.009
PMH of Bleeding Disorders		1.64	1.14	2.25	0.57	0.64	<0.0001
PMH of COPD		3.05	1.81	3.96	0.93	0.80	<0.0001
Recent Pneumonia		0.01	0.00	0.01	0.00	0.00	<0.0001
Dyspnea	At Rest	0.29	0.08	0.42	0.06	0.06	<0.0001
	Moderate	5.56	3.49	7.48	3.60	2.46	
	None	94.15	96.43	92.11	96.33	97.48	
ASA Class	1	6.19	5.19	3.74	6.38	7.88	<0.0001
	2	58.95	60.35	51.60	69.20	69.34	
	3	33.25	33.48	42.22	24.04	21.69	
	4	1.36	0.80	2.33	0.32	0.30	
	5	0.00	0.00	0.00	0.00	0.00	
Open Wound Infection		0.37	0.25	1.31	1.22	0.20	<0.0001
Prior Infection	SIRS	0.20	0.23	0.42	0.55	0.23	<0.0001
(Within 48 h)	Sepsis	0.01	0.01	0.04	0.06	0.00	
Any Operation in Last 30 Days		1.92	0.91	1.30	2.19	0.66	<0.0001
Operating Time	Less than 1 h	0.48	0.16	0.13	0.01	0.01	<0.0001
	1–2 h	0.42	0.37	0.47	0.04	0.11	
	2–3 h	0.08	0.23	0.12	0.07	0.29	
	3–5 h	0.02	0.18	0.02	0.34	0.45	
	5–10 h	0	0.05	0	0.47	0.12	
	10+ h	0	0	0	0.06	0	
Return to OR (Within 30 Days)		0.05	0.04	0.04	0.08	0.07	<0.0001

**Table 3 cancers-11-00253-t003:** Covariates used in Multivariable Logistical Regression.

Surgery Type	PM **, OPS, M, M + R, M + I
Age	Young: <60 Older: >60 **
Race	White **, Black, Asian, Native, Unknown
Pathology	Invasive Breast Cancer Ductal Carcinoma in situ **
Admission Status	Inpatient ** or Outpatient
BMI	Underweight Normal ** OverweightObese
Pregnancy Status	Yes or No **
Smoking Status	Smoker or Non-Smoker **
Smoking Pack Per Day (PPD)	None ** 0–20 PPD 21–50 PPD 50–100 PPD >100 PPD
Steroid Use	Yes or No **
Prior Chemotherapy	Yes or No **
Prior Radiation Therapy	Yes or No **
Peri-Operative ASA Class	1 **, 2, 3, 4, 5
Operative Wound Infection	Yes or No **
Any Operation in Last 30 Days	Yes or No **
Recent Pneumonia	Yes or No **
Diabetic	Non-Diabetic Diabetic on Insulin ** Diabetic on Oral medication
Recent Weight Loss	Yes or No **
PMH of Angina	Yes or No **
PMH of Congestive Heart Failure	Yes or No **
PMH of Renal Failure	Yes or No **
PMH of COPD	Yes or No **
History of Bleeding Disorder	Yes or No **
Operating Time	Less than 1 h ** 1–2 h 2–3 h 3–5 h 5–10 h Over 10 h

* The regression model provides odds ratios compared to a baseline covariate. The asterisks ** is the baseline covariate.

**Table 4 cancers-11-00253-t004:** Complication rate in Surgical interventions.

Categories → Complication (*n*, %)	BCT *n* = 102,747				MAST *n* = 119,456						*p*-Value
	PM *n* = 95,468		OPS *n* = 7279		M *n* = 70,616		M + MF *n =* 4747		M + I *n* = 44,093		
Wound Complication	1341	1.40	128	1.76	2385	3.38	244	5.14	1325	3.01	<0.0001
Infectious	456	0.48	50	0.69	823	1.17	92	1.94	652	1.48	<0.0001
Respiratory	90	0.09	8	0.11	172	0.24	12	0.25	43	0.10	<0.0001
Bleeding	66	0.07	28	0.38	844	1.20	206	4.34	277	0.63	<0.0001
Thromboembolic	99	0.10	9	0.12	200	0.28	55	1.16	179	0.41	<0.0001
Renal	21	0.02	2	0.03	53	0.08	3	0.06	11	0.02	<0.0001
Cardiac	42	0.04	4	0.05	89	0.13	3	0.06	8	0.02	<0.0001
Stroke	29	0.03	4	0.05	64	0.09	4	0.08	10	0.02	<0.0001
Overall Complication Rate	2144	2.25%	233	3.2%	4630	6.56%	619	13.04%	2505	5.68%	<0.0001

BCT. Breast conservation therapy; MAST: Mastectomy group; PM: Partial Mastectomy; OcPs: Oncoplastic Surgery; M: Mastectomy; +MF: Mastectomy with Muscular Flap reconstruction; M + I: Mastectomy with Implant placement.

**Table 5 cancers-11-00253-t005:** Trend of Surgical Complication Rates (*n*, %) *.

Year		Trend of Surgical Complication Rates (*n*, %) *									
		PM		OPS		M		M + I		M + MF	
2005		22	2.7%	1	2.8%	32	5.4%	11	5.5%	4	7.4%
2006		68	2.5%	3	1.6%	112	5.5%	30	5.6%	17	7.4%
2007		103	2.4%	1	2.7%	193	5.6%	73	5.1%	24	8.5%
2008		110	2.3%	7	3.7%	254	5.9%	91	5.0%	38	8.4%
2009		119	2.2%	11	3.7%	326	6.4%	119	5.1%	35	10.7%
2010		115	2.1%	6	3.6%	340	6.9%	147	5.6%	54	13.2%
2011		129	2.1%	15	3.2%	331	7.0%	192	6.0%	64	16.0%
2012		151	2.2%	16	3.1%	359	6.8%	224	5.8%	58	16.5%
2013		177	2.1%	16	2.7%	391	6.6%	249	5.6%	56	16.8%
2014		199	2.3%	21	3.1%	422	6.7%	252	5.5%	54	16.7%
2015		276	2.3%	37	3.1%	436	6.9%	280	5.7%	47	14.7%
2016		278	2.3%	33	3.4%	494	6.8%	320	5.9%	34	13.8%
2017		295	2.2%	45	3.2%	434	6.8%	340	6.1%	41	13.3%
**Overall Trend Analysis**											
*p*-Value	(R^2^)	0.29	(0.27)	0.67	(0.14)	0.004	(0.71)	0.02	(0.39)	0.01	(0.60)

* Percentages are taken from smoothed data analysis to assess trend.

**Table 6 cancers-11-00253-t006:** Multivariable Logistical Regression Analysis of 30-Day Complication Rate: Patient Characteristics and Demographic Predictors.

Complication	OR (95% CI)	Complication	OR (95% CI)
Wound Complications Protective Diabetic on Oral medication Non-Diabetic	0.75 (0.63–0.88) ** 0.61 (0.53–0.70) ***	Infectious Complications Protective Diabetic on Oral medication Non-Diabetic	0.74 (0.56–0.97) * 0.62 (0.49–0.78) ***
Risk		Risk	
PMH of Angina PMH of COPD PMH of Bleeding Disorder Smoker BMI Obese BMI: Overweight	2.22 (1.24–3.98) ** 1.29 (1.09–1.53) ** 1.44 (1.17–1.77) ** 1.61 (1.46–1.77) *** 2.17 (1.97–2.38) *** 1.34 (1.21–1.48) ***	PMH of CHF PMH of COPD PMH of Bleeding Disorder PMH of HTN BMI: Overweight BMI ObeseRecent Weight Loss	2.13 (1.17–3.89) * 1.34 (1.02–1.76) * 1.73 (1.27–2.35) ** 1.20 (1.06–1.37) ** 1.36 (1.16–1.63) ** 1.62 (1.39–1.90) *** 3.48 (2.24–5.42) ***
Respiratory Complications Protective Age <60	0.66 (0.47–0.93) *	Bleeding Complications Protective Non-Diabetic	0.65 (0.49–0.89) **
Risk		Risk	
PMH of COPD BMI: underweight Recent Weight Loss PMH of Bleeding Disorder	1.80 (1.14–2.85) * 2.25 (1.14–4.42) * 2.76 (1.09–6.99) * 2.70 (1.67–4.36) ***	Recent Weight Loss PMH of CHF Steroid Use PMH of Bleeding Disorder	1.96 (1.02–3.79) * 2.97 (1.67–5.26) * 1.82 (1.26–2.62) ** 2.62 (1.90–3.62) ***
Neuro Complications Protective Age <60	0.14 (0.06–0.32) ***	Thromboembolic Complications Risk BMI: Overweight BMI: Obese	1.66 (1.18–2.36) ** 2.45 (1.77–3.40) ***
Renal Complications Protective Non-Diabetic	0.35 (0.18–0.71) **	Cardiac Complications Protective Age <60	0.53 (0.30–0.91) *
Risk		Risk	
PMH of Bleeding Disorder PMH of Renal Failure PMH of CHF Recent Weight Loss	2.50 (1.05–5.94) * 10.45 (1.19–91.74) * 5.06 (1.64–15.59) ** 8.39 (2.87–24.53) **	BMI: Underweight PMH of Angina PMH of HTN	2.94 (1.23–7.03) * 5.20 (1.20–22.46) * 2.34 (1.33–4.12) **

*p*-values: * *p* < 0.05; ** *p* < 0.01–0.001; *** *p* < 0.0001.

**Table 7 cancers-11-00253-t007:** Multivariable logistical Regression Analysis of 30-Day Complication Rate: Surgical Predictors of Morbidity.

Complication	OR (95% CI)	Complication	OR (95% CI)	Complication	OR (95% CI)
Wound Complications		Infectious Complications		Bleeding Complications	
Protective		Protective		Protective	
Outpatient	0.84 (0.77–0.91) ***	Outpatient	0.74 (0.64–0.85) ***	Outpatient	0.20 (0.16–0.6) ***
Risk		Risk		Risk	
M	1.79 (1.62–1.98) ***	M	1.78 (1.48–2.13) ***	OPS	2.78 (1.20–6.45) *
M + MF	2.24 (1.83–2.74) ***	M + MF	2.24 (1.60–3.13) ***	M + I	3.61 (2.28–5.73) ***
M + I	1.72 (1.51–1.96) ***	M + I	2.17 (1.73–2.72) ***	M	5.78 (3.83–8.72) ***
ASA2	1.22 (1.02–1.48) *	ASA2	1.66 (1.14–2.40) **	M + MF	10.99 (6.73–17.90) ***
ASA3	1.57 (1.29–1.91) ***	ASA3	2.44 (1.67–3.58) ***	ASA3	2.18 (1.41–3.37) **
ASA4	1.67 (1.23–2.26) **	ASA4	4.09 (2.50–6.67) ***	ASA4	5.32 (3.12–9.08) ***
Open Wound	1.83 (1.40–2.39) ***	Open Wound	1.98 (1.33–2.94) **	Open Wound	2.09 (1.36–3.21) **
OPT1	1.32 (1.18–1.48) ***	OPT2.	1.36 (1.09–1.71) *	OPT4.	2.92 (1.99–4.28) ***
OPT2	1.51 (1.33–1.72) ***	OPT3.	1.70 (1.34–2.17) ***	OPT5.	7.21 (4.22–12.33) ***
OPT3	1.75 (1.52–2.02) ***	OOPT4.	2.55 (1.90–3.42) ***		
OPT4	2.14 (1.78–2.57) ***	OPT5.	2.64 (1.30–5.37) *		
OPT5	2.81 (1.85–4.27) ***				
Thromboembolic Complications		Cardiac Complications		Respiratory Complications	
Protective		Protective		Protective	
Outpatient	0.70 (0.52–0.95) *	Outpatient	0.20 (0.11–0.35) ***	Outpatient	0.24 (0.16–0.35) ***
Risk		M + I	0.17 (0.04–0.83) *	Risk	
M	2.04 (1.37–3.02) **	OPT1	0.59 (0.35–0.97) *	ASA4	6.62 (2.14–20.48) **
M + MF	3.91 (2.16–7.07) ***	OPT2	0.47 (0.24–0.93) *		
M + I	2.15 (1.33–3.47) **	OPT3	0.28 (0.09–0.86) *		
Prior operation within 30 days	3.20 (1.97–5.20) ***	Risk			
OPT3	1.95 (1.16–3.27) *	ASA4	10.31 (1.23–86.12) *		
OPT4	3.40 (1.89–6.09) ***				
OPT5	6.08 (2.41–15.32) **	Renal Complications		Neuro Complications	
		Protective		Protective	
		Outpatient	0.26 (0.13–0.52) **	Outpatient	0.45 (0.26–0.81) **
				OPT2	0.29 (0.12–0.71) **
				Risk	
				OPS	4.58 (1.28–16.35) *
				M	2.61 (1.31–5.21) **

*p*-values. * *p* < 0.05; ** *p* < 0.01–0.001; *** *p* < 0.0001; ASA2: Class 2 Mild Disease, ASA 3: Class 3 Severe Disease, ASA4: Class 4 Severe Life Threatening; Operative time: OPT1: 1–2 h; OPT2: 2–3 h; OPT:3: 3–5 h; OPT4: 5–10 h; OPT5: >10 h.

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
