# Peer review of "Trending Towards Safer Breast Cancer Surgeries? Examining Acute Complication Rates from A 13-Year NSQIP Analysis"

_cancers, 2019, doi:10.3390/cancers11020253_

Reviewer 1 Report

Timely article given the increasing utilization of breast reconstruction. Findings are not surprising given the extant literature but a) the study power is considerable and b) inclusion of oncoplastic procedures is enlightening. Authors should be applauded for undertaking this study.

Author Response

Dear Reviewer,

Thank you for comments and opportunity reviewing our manuscript, ‘Trending Towards Safer Breast Cancer Surgeries? Examining acute complication rates from a 13-Year NSQIP analysis.’ We greatly appreciate your comments and feedback.

Reviewer 2 Report

Thank you, authors, to perform an extensive, potentially useful analysis of the association among surgery type of either baseline clinical characteristics, and clinical outcome data. 

While this type of data warrants to update the field when clinicians and patients seek data to support their day-to-day decision making, there are some points I would like authors to address as below:

Often times, the decision as to which surgery of breast cancer treatment the patient would go after - is based on the stage, the extent of the disease. this can be associated with age, patient ethnicity, social status, and other factors. I am not sure if the current manuscript has addressed this issue - understanding that retrospective analysis may have some limitation by default. what would authors suggest to address this issue?

More importantly, with these existing biases and confoundings, how can authors conclude the oncoplastic surgery is the best choice - which, from conventional knowledge that selection or the possibility of oncoplastic surgery itself could indicate that cancer itself was much more easily treatable. 

As a minor comment, the abbreviated terms that authors used here certainly can be confusing. for example, OS - is a very common term in the breast cancer field to indicate overall survival. IBC indicates inflammatory breast cancer. I recommend authors to consider overall community acceptable terms to be used in this manuscript.

Author Response

Dear Reviewer 2,

Thank you for comments and opportunity reviewing our manuscript, ‘Trending Towards Safer Breast Cancer Surgeries? Examining acute complication rates from a 13-Year NSQIP analysis.’ The suggestions you offered have been helpful and we appreciate your feedback regarding potential revisions. We feel that the revisions were minor due to a misconstrued terminology, abbreviations and we have addressed the revisions accordingly in the manuscript. Below we provide our feedback addressing each consideration.

Regarding breast cancer intervention, of course this decision is multifactorial. The decision for a particular surgery (BCS or Mast) is influenced by oncological features such as tumor size, staging and we referenced our publication from last year describing how these decisions are multifaceted (Reference #1 Surgical trends in breast cancer: a rise in novel operative treatment options over a 12-year analysis). If oncological feature were available in NSQIP we would have included them in our study but they unfortunately were not. With this being brought to our attention we have added this as a limitation (line 303).  However, tumor size and nodal involvement often influences chemoradiation treatment or axillary management moreso, as BCS and Mast are capable of treating Stages 1-III with intent to cure mentioned by the American Cancer Society. Therefore, our main purpose in this manuscript was to exemplify how comorbidities or surgical prognostic features can influence acute post-operative complications for each surgical intervention. Using other common (we used) demographics age, ethnicity, comorbidities… surgeons to date, have only used few predictors mentioned in line 55-64. Our analysis showed associations and potential predictors, that breast specialists (surgeons) can use in guiding/consenting patients who may be ‘high-risk’ (numerous predictors for morbidity) for post-operative complications (Line 291- 302).

Regarding, biases and confounding, we met with statisticians at Tufts University and after prolific consideration, we decided to use a stepwise statistical approach using covariates mentioned in Table 3 (Line 114). This approach provided an initial screening of covariates removing insignificant or outliers. We then applied them in the adjusted multiple regression model which decreased the probability of confounding. This methodology is problematic in multiple covariates being used with a small sample size leading to potential randomization bias or spurious models. Our cohort involved over 225,000 patients, is very impactful and enhanced the power of our model, hence minimizing those mentioned issues.

Our analysis provided grounds associating preoperative risk factors patients may have and peri-operative variables raising awareness surgeons can acknowledge that may impact a patient’s outcome. In this regard, oncoplastic surgery (OPS) is a rising novel approach to treating breast cancer with a multitude of positive outcomes gaining superiority over other interventions (Trend- Line 39-45; Outcome: Line 241-251). We simply showed associations that other surgical interventions, Mast, had higher likelihood for complication rates. We recommended OPS for this reason in patients with multiple comorbidities seeking a reconstructive procedure and are amendable to OPS. Reviewing the manuscript, our wording was too direct without acknowledging oncologic features. We adjusted our wording appropriately to make a generalizable implication for OPS in conjunction with known publications in our entire paper but also focusing on the discussion/conclusion sections.

Reviewer 3 Report

I think we should rely on the two reviews that we have received: one to accept and one for major revision. 

My recommendation is to inform the authors that the paper would become acceptable should they respond satisfactorily to the concerns raised by reviewer 2.

Author Response

Dear Reviewer,

Thank you for comments and opportunity reviewing our manuscript, ‘Trending Towards Safer Breast Cancer Surgeries? Examining acute complication rates from a 13-Year NSQIP analysis.’ We greatly appreciate all reviewers’ feedback and have made appropriate revisions in addition to a detailed explanation to our conceptualization. We hope the revised manuscript will be better suited for Cancers Special Issue: Treatment Strategies and Survival Outcomes in Breast Cancer. If there are any further considerations we are happy to make the revisions as needed. Thank you for your interest in our research.

Round  2

Reviewer 2 Report

I thank you for authors putting a significant amount of time and perspective into the revision of this manuscript. As a treating physician of patients with breast cancer/abnormalities, I see the importance of this paper that can provide the discussion point for patient-surgeon discussion during daily practice. This paper could be an important reference as the breast surgery techniques and approach evolves.

I do not have an additional recommendation or comment. I recommend this manuscript to proceed to be published upon the final editorial decision.